# Multiple Analysis and Characterization of Novel and Environmentally Friendly Feather Protein-Based Wood Preservatives

**DOI:** 10.3390/polym12010237

**Published:** 2020-01-19

**Authors:** Yan Xia, Chengye Ma, Hanmin Wang, Shaoni Sun, Jialong Wen, Runcang Sun

**Affiliations:** 1Beijing Key Laboratory of Lignocellulosic Chemistry, Beijing Forestry University, Beijing 100083, China; xiayan@swfu.edu.cn (Y.X.); chengye.ma@foxmail.com (C.M.); wanghanmin798@163.com (H.W.); sunshaoni@bjfu.edu.cn (S.S.); 2College of Material Science and Engineering, South-West Forestry University, Kunming 650224, China; 3Center for Lignocellulose Science and Engineering, Dalian Polytechnic University, Dalian 116034, China

**Keywords:** feather protein, wood preservatives, nano-carrier, treatability, decay resistance

## Abstract

In this study, feather was used as the source of protein and combined with copper and boron salts to prepare wood preservatives with nano-hydroxyapatite or nano-graphene oxide as nano-carriers. The treatability of preservative formulations, the changes of chemical structure, micromorphology, crystallinity, thermal properties and chemical composition of wood cell walls during the impregnation and decay experiment were investigated by retention rate of the preservative, Fourier transform infrared spectroscopy (FT-IR), scanning electronic microscopy-energy dispersive spectrometer (SEM-EDS), X-ray diffraction (XRD), thermoanalysis (TG), and confocal Raman microscopy (CRM) techniques. Results revealed that the preservatives (particularly with nano-carrier) successfully penetrated wood blocks, verifying the enhanced effectiveness of protein-based preservative with nano-carrier formulations. Decay experiment demonstrated that the protein-based wood preservative can remarkably improve the decay resistance of the treated wood samples, and it is an effective, environmentally friendly wood preservative. Further analysis of these three preservative groups confirmed the excellent function of nano-hydroxyapatite as a nano-carrier, which can promote the chelation of preservatives with higher content of effective preservatives.

## 1. Introduction

Wood is a conventional construction material, but wood products that have direct contact with outdoor soil without protection easily become less stable and present serious deterioration through the decay and degradation in the ambient environment, which may result huge economic losses and resource waste because wood materials are susceptible to being damaged and destroyed by microorganism such as fungi, bacteria and insects. Based on the above reasons, wooden constructions and architectures are mostly protected and chemically-modified to obtain significant improvements in their stability and durability [1]. In most cases, preservative treatment should be performed on wood products, and the durability and resistance of treated wood products against biological attacks during their service period can be improved [2,3]. Chemical preservatives are common in wood preserving treatments, in which water-soluble preservatives are mostly used. Chromated Copper Arsenate (CCA) preservatives have been the most extensively used in the past decades, especially for wood-framed building timbers. Nevertheless, CCA has been prohibited for residential purposes by the U.S. Environmental Protection Agency since 2004 due to its toxic effect on the environment during manufacture, treatment, and disposal [4,5]. In recent years, some wood preservatives have been subjected to restrictions on its application considering public concern regarding their high toxicity [6,7,8,9]. Based on the above considerations, low toxicity and environmental benign wood preservatives is the research focus in this field, and the development of effective and environment-benign wood preservatives for new preservative systems without chromium and arsenic is necessary [10,11].

In this perspective, copper and boron salts are attracting more and more attention in recent years since they are poisonous to microorganisms and insects but with low toxicity [12,13,14,15,16]. In current circumstances, copper-based preservatives, such as Ammonium Copper Quaternary (ACQ) and Chromated Copper Arsenate (CCA) are the most common preservatives. Since some species of fungi can develop resistance to copper salts, copper-based preservatives are usually applied with other active ingredients to achieve a better preservative effect. Boron-based preservatives are also commonly used because of their low toxicity, resistance against fungi and insects, and low-cost characteristics. In contrast, borates are not suitable for outdoor application because they are easily to leach out due to borates’ preferable water-soluble property. To avoid the leaching of active ingredients from preservative caused by their water solubility, proteins can be used as fixative agents, such as soy isolates and egg albumin, to chelate boron and copper in the preservative by chelation, coagulation, and/or chemical reactions to form insoluble complexes thus increase the fixation and durability of preservatives during the wood treating process [17,18,19,20,21,22,23]. Nevertheless, since protein is a kind of nutritious matter, excessive amounts of protein may precipitate in wood blocks. This suggests that the excessive protein can serve as a nutrient source consumed by fungi, which might cause a loss of Cu and B in the decay process. Therefore, the ideal protein-based wood preservative should chelate more preservative components and contain less protein.

Based on this, many researchers seek alternative solutions to solve the problem about preservative fixation. Furthermore, to promote the penetration depth and uniformity of the active preservative components, nano-carriers are frequently used in the preparation of wood preservative to enhance the content of effective preservative ingredients [24,25,26,27,28]. In this study, feather protein-based preservatives with different nano-carriers were firstly developed. The treatability of preservative formulations, the changes of chemical structure, micromorphology, crystallinity, thermal properties, and chemical composition of wood cell walls during the modification and decay test were comprehensively investigated. Results show that the feather protein-based preservative studied in this paper can significantly improve the resistance performance against decay fungi of the treated wood, and it is believed that the protein-based preservatives with different nano-carriers have great potential in the fabrication of eco-friendly wood products.

## 2. Materials and Methods

### 2.1. Preparation of Feather Protein-Based Wood Preservatives

The preservative formulations were made from hydrolyzed feather protein, copper sulfate (CuSO_4_·5H_2_O) and sodium borate (Na_2_B_4_O_7_·10H_2_O). Protein hydrolysate was obtained by hydrolyzing chicken feather powder at 140 °C for 4 h after immersed in 6 *wt*% aqueous sodium hydroxide at room temperature for 24 h. The concentration of hydrolyzate was condensed to 50% and it was added into the suspension of copper sulfate and sodium borate with the ratio of protein to total amounts of Cu and B in the formulations of 1:1, *w*/*w*. Then, a few drops of glacial acetic acid were added into the mixture. Commercial ammonium hydroxide (NH_4_OH) with a one-tenth volume of the suspension was added to dissolve the water-insoluble mixture and obtain preservative solution, which was named feather protein-based wood preservative in this study.

To further increase the performance of the preservative, nano-hydroxyapatite or nano-graphene oxide was added into feather protein-based preservative as nano-carriers and blended by the ultrasonic vibration, which facilitates the uniform distribution of these nano-particles in the newly developed preservative. In this study, there are three preservative formulations (P_1_, P_2_ and P_3_). P_1_ was prepared by feather protein combined with copper sulfate and sodium borate, accordingly named the feather protein-based preservative (Cu-B-Pr). Preservative P_2_ and P_3_ were prepared by the feather protein combined with copper sulfate and sodium borate and nano-hydroxyapatite, nano-graphene oxide, respectively. Based on these formulations, P_2_ and P_3_ were correspondingly named nano-hydroxyapatite protein-based preservative, and the nano-graphene oxide protein-based preservative (Cu-B-Pr-HA, Cu-B-Pr-Go). The preparation procedure for the protein-based preservative was shown below (Scheme 1).

### 2.2. Preservative Treatment of the Wood Blocks 

Wood blocks sawed from *Pinus* yunnanensis sapwood with a dimension of 150.0 × 20.0 × 20.0 mm (size in axial, radial, and tangential) were each immersed in preservative formulation for 24 h at normal temperature and pressure conditions, and then oven-dried at 60 °C for 24 h, followed by air-dried over 24 h. Each preservative formulation with size of 10.0 × 20.0 × 20.0 mm (size in axial, radial, and tangential) was used for determining decay experiments.

### 2.3. Treatability of the Preservatives

To measure the solution uptake of treated samples in preservative formulations, treated wood blocks were air conditioned for 24 h and then oven-dried at 60 °C for 24 h. Treatability, representing actual percent retention of the preservatives in the treated samples, was calculated through the ratio of measured retention and target retention for the preservative.

### 2.4. Decay Resistance of Treated Wood Samples

Decay resistance of control and the treated wood blocks exposed to decay fungi was evaluated according to the method described in ASTM Standard D 1413-07. Brown-rot fungi *Gloeophyllum trabeum* (GT) was used as the test fungi in decay experiments [29].

Fungus cultured on potato dextrose agar was inoculated on the feeder strips on the surface of a mixture composed of river sand, sawdust, corn flour and brown sugar. After the fungal mycelia covered the surface, sterilized wood blocks were placed onto the feed strips, two blocks per bottle. Culture bottles were sterilized for 1 h before being inserted into the decay chamber. The soil-block culture was incubated at 26 ± 1°C and 75% relative humidity for 12 weeks. After the incubation, wood blocks were moved out from the culture bottles in the decay chambers, and the fungal mycelia were cleaned, then dried overnight in an oven at 80 °C and weighed to determine weight loss of the wood samples. The decay rate of the wood block was represented by the percentage weight loss during exposure to the decay fungus. The treated wood blocks (24 pieces of wood) with different preservatives formulations were evaluated by the decay test.

### 2.5. Multi-Analysis of Control and Treated Samples

To better investigate the effect of preservative treatment on the treatability of protein-based preservatives, multi-analysis of control and treated samples (12 weeks decay test) was conducted to investigate the decay resistance performances against fungi of the three preservative formulations.

Fourier transform infrared (FT-IR) spectra analysis was used on Thermo Scientific Nicolet iN10 FT-IR microscope (Thermo Nicolet Corporation, Waltham, MA, USA), which was conducted in the range from 4000 to 400 cm^−1^ with 64 scansions per sample at a resolution of 4 cm^−1^ [30,31]. Control and wood samples were milled into powders (40–60 mesh) and then analyzed to elucidate the changes before and after the treatment with preservatives.

Crystallinity index was measured by X-ray diffraction (XRD) using Ni-filtered CuKa radiation at 40 kV and 30 mA. The crystallinities of the specimens were calculated by the ratio of areas under crystalline peaks and amorphous curve according to previous publications [32,33]. Thermogravimetric analysis (TGA) was performed on a simultaneous thermal analyzer DTG-60 (Shimadzu, Kyoto, Japan). 3-5 mg samples were heated in an alumina crucible at a heating rate of 10 °C·min^−1^ from room temperature to 600 °C under nitrogen atmosphere [34]. 

Scanning Electron Microscope (SEM) can clearly observe the micromorphology and microstructure change of plant cell walls. Energy Dispersive Spectrometer (EDS) can analyze the chemical composition of the cell walls of controlled and treated samples. SEM images were executed with a Hitachi S-3400 N II (Hitachi, Tokyo, Japan) instrument at 10 kV and 81 mA [35]. 

Raman spectra were acquired with a confocal Raman microscope (CRM, LabRam Xplora, HORIBA, Kyoto, Japan), which was equipped with a piezo scanner and a high numerical aperture (NA) microscope objective from Olympus (100oil NA = 1.40). The Labspect5 software (HORIBA) was used for measurement setup and image processing to remove spikes, smooth the spectra by the Savitsky-Golay algorithm at a moderate level, correct baselines, and the data was further smoothed by Fourier transformation coupled with cosine apodization function [34,35,36]. Cross sections of 10 μm thickness were cut from wood sample using a rotary microtome RM 2255 (Leica, Wetzlar, Germany) to obtain a full wafer and then covered with glass cover slips. The chemical images allowed us to separate cell wall layers into secondary wall (S) and the cellular corner middle layer (CCML) with different chemical compositions, and to mark distinct cell wall regions for constructing average spectra.

## 3. Results and Discussion

### 3.1. Treatability of Protein-Based Preservatives

Treatability of preservative formulations means actual percent retention of the protein-based preservatives in treated wood blocks, which was listed in Table 1. As shown in Table 1, the measured retentions of Cu and B in wood samples treated with P_1_, P_2_ and P_3_ were very close to the target retention, respectively. The treatability of Cu in pretreated wood samples was 84.6% to 87.9%, while the treatability of B in pretreated wood samples was 89.6% to 95.5%. This fact suggested that the three preservative formulations could effectively penetrate wood blocks since ammonium hydroxide is a good dissociating agent [4,5].

### 3.2. FT-IR Analysis before/after Decay

To compare the structural changes of wood samples after the preservative treatment, the fingerprint region in the FT-IR spectra of control and treated wood samples are presented in Figure 1. As can be seen from Figure 1, it was found that the spectra at 3350 and 2900 cm^−1^ decreased distinctly, revealing that a relatively high content of the hydroxy and aliphatic acid extractives could interact with preservative ingredients during the treatment process. It was observed that the signals at 1740 cm^−1^ for hemicellulose almost disappeared, suggesting that deacetylation of hemicelluloses occurred during the impregnation stage. The absorption bands at 1590 cm^−1^, 1505 cm^−1^ attributed to aromatic skeletal vibration breathing with C=O stretching in the lignin fraction signals, were observed to be weakened. Moreover, the peaks at 1460 cm^−1^ and 1370 cm^−1^ corresponding to cellulose, hemicellulose and lignin, are also significantly diminished or decreased sharply, implying that effective interaction occurred between cellulose, hemicellulose and lignin with three groups of preservative.

After a decay resistance test, the absorption peaks at 3350, 2900, 1740, 1160, 1040 cm^−1^ are distinctly weakened in the spectra of control sample, indicating that cellulose and hemicelluloses were partly destroyed during the decay process. By contrast, the main components in the treated samples remained relatively steady after the decay test, suggesting that treated wood samples have been effectively protected. In short, the data presented herein revealed that the protein-based preservative systems were effective formulations and constituted appropriate protections for treated wood blocks.

### 3.3. Morphology Analysis

The morphology of control and treated samples was investigated by SEM, and the distributions and contents of Cu and B within the wood cell wall were also analyzed by SEM-EDS, as shown in Figure 2. To observe the cross-section morphology by SEM, a small piece was randomly cut from the inside of the wood blocks, after removing at least 5 mm from the edge, completed with the preservative modification treatment and wood drying. The cross-section and radial-section morphologies of the control and treated wood samples are shown in Figure 3. From the magnified images, deposition of Cu-B-protein was not observed inside tracheid in these samples.

SEM pictures of control and treated samples with protein-based wood preservatives demonstrated that Cu and B elements have successfully impregnated and penetrated uniformly into the treated wood cell walls. SEM-EDS analysis showed the distributions and contents of active preservative elements within the cell walls before and after decay tests, and the results are shown in Table 2. It was found that elements (Cu and B) distributions in the samples treated by P_2_ and P_3_ preservatives were higher than those in P_1_ treated wood samples. In contrast, lower N contents were observed in P_2_ and P_3_ treated wood samples. These results showed that the P_2_, P_3_ preservatives with nano-carriers can chelate more active components (Cu, B) and less protein in preservative formulations, facilitating the permeation of preservatives into the treated wood cell walls. In particular, for the samples treated by P_2_ preservative with nano-hydroxyapatite, the protein content in cell walls was the least but Cu and B contents were at high levels. Since protein is also a kind of nutritious matter, preferable wood preservative should chelate less protein and more active components for protecting wood materials from decay. Consequently, it can be concluded that nano-carriers in P_2_ preservatives could promote more active ingredients permeating into treated wood cells. 

After 12 weeks decay test, the entire cell wall can scarcely be found in the control samples (Figure 4), suggesting that serious degradation of cell walls had occurred after the decay process. In contrast, the cell walls were relatively unchanged in the three treated wood samples, further suggesting the effectiveness of protein-based preservatives. The cell wall treated by preservative P_2_ is more intact than samples modified by preservative P_1_ and P_3_. This might be attributed to the high contents of Cu and B but with less protein content of P_2_. In general, the morphology results reflected by SEM-EDS and SEM pictures confirmed that the active ingredients in preservatives can effectively penetrate and fixate within the wood blocks, especially in the P_2_ preservative.

### 3.4. XRD Analysis

In this study, the crystallinity index (C_r_I) of control and treated samples was measured by XRD-6000 instrument (Shimadzu, Japan). The X-ray diffraction patterns of wood samples are presented in Figure 5. The C_r_I values were calculated and shown in Table 3. As can be seen from Figure 5, the XRD pattern of wood samples all showed typical cellulose I structure, indicating that the crystal structures of the treated samples were not changed by preservatives during the impregnation processes. The unchanged crystal structures are beneficial for utilization of the modified wood, and contributed to some properties, such as strength of treated wood blocks, remaining unchanged.

As can be seen from the Table 3, the C_r_I of preservative-treated wood samples all reduced (from 50.66 to 51.27%) distinctly as compared to that (60.1%) of control sample, which might be attributed to the addition of amorphous protein in the preservative. The CrI (50.69% and 50.66%) values of treated wood samples with the preservatives P_1_, P_3_ were less than that treated with P_2_ (51.27%). This might attribute to the function of nano-carrier in P_2_ preservative, which promotes more active ingredients chelating less protein and facilitates preservatives into the wood cells.

After 12 weeks of decay tests, the C_r_I of three treated wood samples all substantially decreased (44.66%–46.92%), whereas the C_r_I of control samples decreased to 40.33%. Fungi can directly attack and degrade the main chemical components of the control wood blocks during the decay process, such as cellulose and hemicelluloses. The CrI of P_2_ (46.92%) treated samples was slightly higher than those of P_1_, P_3_ treated samples (44.66% and 45.49%) after decay, which was attributable to the function of nano-carrier in preservatives. P_2_ preservative contained a high content of active ingredients and low protein content and it showed more effective protection for wood materials, demonstrating that P2 is the most effective preservative.

### 3.5. Thermogravimetric Analysis (TGA)

TGA was used to evaluate the thermal properties of control and modified wood samples. As shown in Figure 6 (before the decay process), the initial pre-carbonization temperature of preservative-treated samples was lower than that of the control samples. This might attribute to the facilitation for char forming derive from Cu and B elements. The final carbon residue of the treated samples (P_1_, P_2_ and P_3_) was considerably higher than control sample since preservative ingredients can promote the carbonization and retard the thermal decomposition of the wood components.

As can be seen from Figure 6 (after the decay process), the content of residual char in the control sample increased as compared to that before decay test. The increased content of “char residues” in the control sample is likely attributed to the high content of lignin in the decayed sample, which is due to the serious degradation of cellulose and hemicelluloses during the decay stage. In contrast, the content of residual char in the treated samples (P_1_, P_2_ and P_3_) after the decay test was close to that before the decay test, demonstrating that preservatives effectively inhibit wood decay, and the loss of treated wood is not obvious after decay treatment. Furthermore, the content of “char residues” in P_2_ and P_3_ treated wood samples is higher than that of P_1_ without nano-carrier. This indicated that wood products pretreated by P_2_ and P_3_ preservatives have higher thermostability, which can extend the application range of wood products in different conditions.

### 3.6. Raman Analysis

Raman analysis was performed to reveal the distribution and microscopic changes of the main structural compositions at subcellular level. The morphological and compositional information of control and the treated samples were simultaneously recorded by the CRM, and the intensity of the bands may be used for the calculation of the relative content in the samples [36,37,38]. Obvious differences between three treated groups and the control sample can be observed in Figure 7 before the decay test. The changes in the contents of carbohydrates and lignin in the wood cell wall implied that the three kinds of preservative (P_1_, P_2_ and P_3_) all can penetrate into the treated wood cell walls.

As observed in Raman spectra, the control wood cell wall was intact before the decay process, and the concentration of carbohydrates was high in the S_2_ layer. After the decay stage, it was found that the distribution of carbohydrates and lignin decreased significantly. However, the cell walls of three treated samples (P_1_, P_2_, and P_3_) were ultimately well preserved after the decay test, although the decrease of carbohydrates and lignin distribution were observed, implying that the preservative can protect the wood from degradation.

It can be seen from Figure 7, after the decay stage, that the concentrations of carbohydrates and lignin in the P_2_ treated cell wall are higher than those in P_1_ and P_3_, suggesting that the P_2_ treated cell wall remained relatively intact. This might be attributed to the nano-carrier function in the P_2_ preservative, which could promote high contents of effective ingredients chelating within the preservative formulations.

Verification was further performed by Raman spectroscopy, as shown in Appendix A. In Raman spectra, the average signal intensity in the spectral ranges of 1550–1650 cm^−1^ and 2880–2920 cm^−1^ are respectively applied to assess the carbohydrates and lignin distributions. From Appendix A, an obvious trend can be found, which can be attributed to the distinct decreases of peak intensity generated by the huge reduction in carbohydrates and lignin for the control sample after the decay process. Compared to the control sample, the intensities of the carbohydrates and lignin signals in the treated samples were also reduced, indicating the relative degradation of cell walls treated by three formulation preservatives (P_1_, P_2_ and P_3_). The reduction in peak intensities of carbohydrates in P_2_ was the least in the three treated samples, further indicating the better protection effectiveness of P_2_ formulation.

In short, Raman analysis demonstrated that the ingredients of preservatives can effectively impregnate into the wood cell wall and adequately protect the treated wood blocks. Furthermore, P_2_ preservative formulation is the most optimal one in the three preservative formulations, which is consistent with the aforementioned SEM, XRD analysis.

### 3.7. Mass Loss Analysis after Decay Test

As can be seen from Table 4 and Figure 8, the mass loss rates of the control sample were significantly much higher than those of the treated samples, indicating that the treated samples (P_1_, P_2_ and P_3_) exhibited strong resistance against decay fungi. It was illustrated that these three formulation preservatives can effectively protect wood blocks. In the treated wood samples, the mass loss rate (9.1%) of sample treated with P_2_ formulation was the lowest, suggesting that P_2_ exhibited the optimal preservative effect, which might be attributed to the nano-carrier function of nano-hydroxyapatite (HA). It can chelate more content of active ingredients and fix the preservative ingredient (Cu and B) into the cell walls of wood, which is in accordance with the results. In addition, weight percent gain (WPG) of P_1_, P_2_ and P_3_ was 20.5%, 20.3% and 22.0%, respectively. The least mass loss rate and weight percent gain indicated that the P_2_ can better protect the wood products from degradation.

## 4. Conclusions

In this study, the results indicated that protein-based preservatives could serve as effective, environmentally friendly and cost-competitive alternatives for traditional wood preservatives. In this formulation system, copper and boron salts are preferably fixed together and exhibit a durable performance on account of the feather protein being introduced as a chelation agent to form insoluble complexes by chelation, instigate chemical reactions with wood components and form a long-term protection mechanism in woodblocks. This enables the feather protein-based preservative to be fairly appropriate for wood construction. Treatability and morphology of the control and treated samples further verified the excellent permeability and feasibility of protein-based preservative formulations. SEM-EDS and Raman analysis of the control and treated samples after decay experiments illustrated the good performance of nano-carriers for the Cu, B penetration and fixation of the protein-based preservative. In particular, the nano-hydroxyapatite preservative formulation could increase the content of Cu and B in preservative at low protein levels. In the future, the protein-based preservatives with nano-carrier (nano-hydroxyapatite) should be further evaluated by field trials to identify their long-term ground-contact applicability.

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
