# Peer review of "Multiple Analysis and Characterization of Novel and Environmentally Friendly Feather Protein-Based Wood Preservatives"

_polymers, 2020, doi:10.3390/polym12010237_

Round 1
Reviewer 1 Report
The author prepared and used non toxic preservative applicable in wood, its an interesting article and written well.
Before publication some minor revisions needed
It would be easier to reader to understand the preparation if the preservative preparation can be summarized in scheme. Introduction is too short, please add more about the current preservative in wood application. Please add error bar in Figure 8
Reviewer 2 Report
Title: Multiple analysis and characterization of a novel and environmentally friendly feather protein-based wood preservatives
This study investigated the feasibility of using feather as the source of protein and combining with copper and boron salts to prepare wood preservatives, with nano-hydroxyapatite or nano-graphene oxide as nano-carriers. The treatability of preservative formulations, the changes of chemical structure, micromorphology, crystallinity, thermal properties and chemical composition of wood cell walls during the impregnation and decay experiment were investigated by retention rate of the preservative, Fourier transform infrared spectroscopy, scanning electronic microscopy-energy dispersive spectrometer, X-ray diffraction, thermoanalysis, and confocal Raman microscopy techniques. The paper is interesting in general and falls into the scope of journal.
Please consider the following comments and suggestions.
1 In the Introduction part, are there any other solutions for wood preservative? A brief summary upon the pros and cons of other ways should be given. Also, wood is one of sustainable construction materials and it is getting more and more popular. How about possible ways to retrofit the deteriorated wood in construction field? Can the current work benefit the wood materials and promote its durability when wood is used in construction field? It will be more interesting to give a brief explanation.
2 In the introduction part, what is real current knowledge gap compared to a lot of other preservation approaches in this field? What is the novelty of current work compared to previous? Please clarify. Some of the related literatures are recommended for the authors to refer as follows.
[1] Potential of teak heartwood extracts as a natural wood preservative. Journal of Cleaner Production, 2017, 142: 2093-2099.
[2] A review of wood-frame low-rise building performance study under hurricane winds. Engineering Structures, 2017, 141: 512-529.
[3] Effect of moisture on the mechanical properties of CFRP–wood composite: an experimental and atomistic investigation. Composites Part B: Engineering, 2015, 71: 63-73.
3 The lines in figure 2 are a bit hard to distinguish. Modification is needed.
4 According to the conclusion, it is claimed that protein-based preservatives could be served as effective, environmentally friendly and cost-competitive alternatives for traditional wood preservatives. Is there any comparison upon cost evaluation of current way with other approaches? A solid backup for the claim is needed.
Round 2
Reviewer 2 Report
Can be accepted after correcting the title and author names of all references. There are some errors in some references.
Author Response
Zhou, A.; Tam, L.-h.; Yu, Z.; Lau, D. Effect of moisture on the mechanical properties of CFRP–wood composite: an experimental and atomistic investigation. Composites Part B: Engineering 2015, 71, 63-73. Zhu, Y.; Zhuang, L.; Goodell, B.; Cao, J.; Mahaney, J. Iron sequestration in brown-rot fungi by oxalate and the production of reactive oxygen species (ROS). International Biodeterioration & Biodegradation 2016, 109, 185-190. Can, A.; Sivrikaya, H.; Hazer, B. Fungal inhibition and chemical characterization of wood treated with novel polystyrene-soybean oil copolymer containing silver nanoparticles. International Biodeterioration & Biodegradation 2018, 133, 210-215. Yang, I.; Kuo, M.; Myers, D.J. Soy protein combined with copper and boron compounds for providing effective wood preservation. Journal of the American Oil Chemists’ Society 2006, 83, 239. Ahn, S.H.; Oh, S.C.; Choi, I.-G.; Kim, H.-Y.; Yang, I. Efficacy of wood preservatives formulated from okara with copper and/or boron salts. Journal of Wood Science 2008, 54, 495-501. Yamaguchi, H. Low molecular weight silicic acid–inorganic compound complex as wood preservative. Wood Science and Technology 2002, 36, 399-417. Humar, M.; Lesar, B. Influence of dipping time on uptake of preservative solution, adsorption, penetration and fixation of copper-ethanolamine based wood preservatives. European Journal of Wood and Wood Products 2009, 67, 265-270. Kim, H.Y.; Jeong, H.S.; Min, B.C.; Ahn, S.H.; Oh, S.C.; Yoon, Y.h.; Choi, I.G.; Yang, I. Antifungal efficacy of environmentally friendly wood preservatives formulated with enzymatic‐hydrolyzed okara, copper, or boron salts. Environmental Toxicology and Chemistry 2011, 30, 1297-1305. Murguía, M.C.; Machuca, L.M.; Fernandez, M.E. Cationic gemini compounds with antifungal activity and wood preservation potentiality. Journal of Industrial and Engineering Chemistry 2019, 72, 170-177. Thevenon, M.-F.; Pizzi, A.; Haluk, J.-P. Protein borates as non-toxic, long-term, wide-spectrum, ground-contact wood preservatives. Holzforschung-International Journal of the Biology, Chemistry, Physics and Technology of Wood 1998, 52, 241-248. Brocco, V.F.; Paes, J.B.; da Costa, L.G.; Brazolin, S.; Arantes, M.D.C. Potential of teak heartwood extracts as a natural wood preservative. Journal of Cleaner Production 2017, 142, 2093-2099. Lesar, B.; Kralj, P.; Humar, M. Montan wax improves performance of boron-based wood preservatives. International Biodeterioration & Biodegradation 2009, 63, 306-310. Ramos, A.; Jorge, F.C.; Botelho, C. Boron fixation in wood: studies of fixation mechanisms using model compounds and maritime pine. Holz als Roh-und Werkstoff 2006, 64, 445. Thevenon, M.-F.; Pizzi, A.; Haluk, J.-P. Non-toxic albumin and soja protein borates as ground-contact wood preservatives. Holz als Roh-und Werkstoff 1997, 55, 293-296. Lebow, S.; Arango, R.; Woodward, B.; Lebow, P.; Ohno, K. Efficacy of alternatives to zinc naphthenate for dip treatment of wood packaging materials. International Biodeterioration & Biodegradation 2015, 104, 371-376. Toussaint-Dauvergne, E.; Soulounganga, P.; Gerardin, P.; Loubinoux, B. Glycerol/glyoxal: a new boron fixation system for wood preservation and dimensional stabilization. Holzforschung 2000, 54, 123-126. Mourant, D.; Yang, D.-Q.; Lu, X.; Riedl, B.; Roy, C. Copper and boron fixation in wood by pyrolytic resins. Bioresource Technology 2009, 100, 1442-1449. Ahn, S.H.; Oh, S.C.; Choi, I.-g.; Han, G.-s.; Jeong, H.-s.; Kim, K.-w.; Yoon, Y.-h.; Yang, I. Environmentally friendly wood preservatives formulated with enzymatic-hydrolyzed okara, copper and/or boron salts. Journal of Hazardous Materials 2010, 178, 604-611. Thevenon, M.-F.; Pizzi, A.; Haluk, J.-P. One-step tannin fixation of non-toxic protein borates wood preservatives. Holz als Roh-und Werkstoff 1998, 56, 90-90. Thévenon, M.-F.; Pizzi, A.; Haluk, J.; Zaremski, A. Normalised biological tests of protein borates wood preservatives. European Journal of Wood and Wood Products 1998, 56, 162-162. Thévenon, M.-F.; Pizzi, A. Polyborate ions’ influence on the durability of wood treated with non-toxic protein borate preservatives. Holz als Roh-und Werkstoff 2003, 61, 457-464. Ratajczak, I.; Mazela, B. The boron fixation to the cellulose, lignin and wood matrix through its reaction with protein. Holz als Roh-und Werkstoff 2007, 65, 231. Mazela, B.; Domagalski, P.; Mamonova, M.; Ratajczak, I. Protein impact on the capability of the protein-borate preservative penetration and distribution into pine and aspen wood. Holz als Roh-und Werkstoff 2007, 65, 137. Lykidis, C.; Mantanis, G.; Adamopoulos, S.; Kalafata, K.; Arabatzis, I. Effects of nano-sized zinc oxide and zinc borate impregnation on brown rot resistance of black pine (Pinus nigra L.) wood. Wood Material Science & Engineering 2013, 8, 242-244. Moradi, F.G.; Hejazi, M.J.; Hamishehkar, H.; Enayati, A.A. Co-encapsulation of imidacloprid and lambda-cyhalothrin using biocompatible nanocarriers: Characterization and application. Ecotoxicology and Environmental Safety 2019, 175, 155-163. Yusoff, S.; Kamari, A.; Aljafree, N. A review of materials used as carrier agents in pesticide formulations. International Journal of Environmental Science and Technology 2016, 13, 2977-2994. Munir, M.U.; Ihsan, A.; Sarwar, Y.; Bajwa, S.Z.; Bano, K.; Tehseen, B.; Zeb, N.; Hussain, I.; Ansari, M.T.; Saeed, M. Hollow mesoporous hydroxyapatite nanostructures; smart nanocarriers with high drug loading and controlled releasing features. International Journal of Pharmaceutics 2018, 544, 112-120. Gholibegloo, E.; Karbasi, A.; Pourhajibagher, M.; Chiniforush, N.; Ramazani, A.; Akbari, T.; Bahador, A.; Khoobi, M. Carnosine-graphene oxide conjugates decorated with hydroxyapatite as promising nanocarrier for ICG loading with enhanced antibacterial effects in photodynamic therapy against Streptococcus mutans. Journal of Photochemistry and Photobiology B: Biology 2018, 181, 14-22. Edmunds, C.W. Physico-chemical properties and biodegradability of genetically modified Populus trichocarpa and Pinus taeda. 2015. Sun, S.-L.; Wen, J.-L.; Ma, M.-G.; Song, X.-L.; Sun, R.-C. Integrated biorefinery based on hydrothermal and alkaline treatments: investigation of sorghum hemicelluloses. Carbohydrate Polymers 2014, 111, 663-669. You, T.-T.; Zhang, L.-M.; Zhou, S.-K.; Xu, F. Structural elucidation of lignin–carbohydrate complex (LCC) preparations and lignin from Arundo donax Linn. Industrial Crops and Products 2015, 71, 65-74. Chen, J.-H.; Guan, Y.; Wang, K.; Xu, F.; Sun, R.-C. Regulating effect of hemicelluloses on the preparation and properties of composite Lyocell fibers. Cellulose 2015, 22, 1505-1516. Chen, J.H.; Guan, Y.; Wang, K.; Xu, F.; Sun, R.C. Regenerated cellulose fibers prepared from wheat straw with different solvents. Macromolecular Materials and Engineering 2015, 300, 793-801. Yuan, T.-Q.; Zhang, L.-M.; Xu, F.; Sun, R.-C. Enhanced photostability and thermal stability of wood by benzoylation in an ionic liquid system. Industrial Crops and Products 2013, 45, 36-43. Nguyen, T.T.H.; Li, S.; Li, J.; Liang, T. Micro-distribution and fixation of a rosin-based micronized-copper preservative in poplar wood. International Biodeterioration & Biodegradation 2013, 83, 63-70. Li, H.-Y.; Sun, S.-N.; Wang, C.-Z.; Sun, R.-C. Structural and dynamic changes of lignin in Eucalyptus cell walls during successive alkaline ethanol treatments. Industrial Crops and Products 2015, 74, 200-208. Zhang, X.; Ma, J.; Ji, Z.; Yang, G.H.; Zhou, X.; Xu, F. Using confocal Raman microscopy to real‐time monitor poplar cell wall swelling and dissolution during ionic liquid pretreatment. Microscopy Research and Technique 2014, 77, 609-618. Zhou, X.; Ma, J.; Ji, Z.; Zhang, X.; Ramaswamy, S.; Xu, F.; Sun, R.C. Dilute acid pretreatment differentially affects the compositional and architectural features of Pinus bungeana Zucc. compression and opposite wood tracheid walls. Industrial Crops and Products 2014, 62, 196-203.